# Prediction of Received Optical Power for Switching Hybrid FSO/RF System

**Renát Haluška \*** , **Peter Šuľaj, Ľuboš Ovseník \*, Stanislav Marchevský, Ján Papaj** and **Ľubomír Doboš**

Department of Electronics and Multimedia Communications, Technical University of Košice, 04000 Košice, Slovakia; peter.sulaj@tuke.sk (P.Š.); stanislav.marchevsky@tuke.sk (S.M.); jan.papaj@tuke.sk (J.P.); lubomir.dobos@tuke.sk (Ľ.D.)

**\*** Correspondence: renat.haluska@tuke.sk (R.H.); lubos.ovsenik@tuke.sk (Ľ.O.); Tel.: +42-1(55)-602-2865 (R.H.)

**Abstract:** This study deals with the problem of fiber-free optical communication systems—known as free space optics—using received signal strength identifier (RSSI) prediction analysis for hard switching of optical fiber-free link to base radio-frequency (RF) link and back. Adverse influences affecting the atmospheric transmission channel significantly impair optical communications, therefore attention was paid to the practical design, as well as to the implementation of the monitoring device that is used to record and process weather information along a transmission path. The article contains an analysis and methodology of the solution of the high availability of the optical link. Attention was paid to the technique of hard free space optics (FSO)/RF-switching with regard to the amount of received optical power detected and its relation to the quantities influencing the optical communication line. For this purpose, selected methods of machine learning were used, which serve to predict the received optical power. The process of analysis of prediction of received optical power is realized by regression models. The study presents the design of the optimal data input matrix model, which forms the basis for the training of the prediction models for estimating the received optical power.

**Keywords:** availability; FSO; hybrid FSO/RF; machine learning

## 1. Introduction

Optical fiber-free communication systems, known as free space optics (FSO) systems, provide high-speed point-to-point communication. It is characterized by high transmission speeds over distances in units of kilometers. The wavelengths of FSO lines fall within the unlicensed frequency spectrum. The use of FSO systems has found its use in solving the so-called last mile problem. Because of the necessity of ensuring comparable transmission speed with fiber optics, it is possible to deploy the FSO line to cover the last few kilometers or hundreds of meters, respectively, to the final goal. The financial comparison of the application of the standard solution (the placement of the optical fiber) and the alternative approach with the help of the non-wearable optics shows the undeniable advantages of FSO systems [1].

The drawbacks of FSO systems include the throughput of the atmospheric channel, which is strongly dependent on current weather conditions between the FSO transmitter and receiver. The need for direct visibility between FSO devices is limited by the maximum possible distance between the communication link of the FSO line. The radiated optical signal naturally diverges (the optical divergence effect) and there is a loss in the received optical power on the receiver side. Using wavelengths less than 800 nm may result in serious damage to human vision [2,3].

The mutual interaction of the signal from the FSO heads (optical beam) with aerosols leads to a deterioration of the quality of communication in the atmosphere, a reduction of the transmission speed—and in the worst case—to a complete interruption of communication. From a practical point of view, this means that the amount of received optical signal detected on the photo-sensitive element, e.g., on the avalanche photodiode (APD) the FSO head decreases depending on the number of scattered particles in the atmosphere along the transmission channel [4]. The size and concentration of the fog particles are the biggest problem for the FSO signal, as they cause critical values of the attenuation of the transmitted optical signal [5]. By suitable identification and prediction of the occurrence of unfavorable weather conditions, it is possible to switch the FSO line to a secondary radio-frequency (RF) line. The RF line is less sensitive to the presence of fog, but on the other hand, RF communication is degraded by rain. By solving the question of the application of models of the development of physical quantities identifying the atmospheric channel with respect to the received optical power (quality and intensity of the received signal) it is possible e to ensure efficient switching between the primary FSO and the backup RF line [6,7]. The aim is to design a model that would use machine-learning methods (random forest, gradient-boosting regression, decision-tree regression) to reliably predict and identify the level of received optical power, thereby increasing the availability and reliability of free space optics/radio frequency (FSO/RF) systems. The method of random forests is based on the principle of use methods of decision-trees, which are further divided into classification Moreover, regression trees (CART). The advantage of classification and regression trees is their simple graphic interpretation. The drawbacks of decision-trees in general is their susceptibility to small changes in data, which results in different outputs [8].

The received signal strength identifier (RSSI) parameter is necessary in the hard-switching process used in the hybrid FSO/RF system. It deals with the design and implementation of obtaining information on RSSI value in real time, which serves as an indicator of the state when the FSO system is in the state before full outage of communication. Visibility is a very important prerequisite for the operation of FSO systems and is characterized by the transparency of the atmosphere estimated by the observers. The current visibility value is a useful indicator characterizing atmospheric conditions with respect to the occurrence of various types of particles present in the atmosphere (fog, haze, clouds and other particles). The presence of a thick fog between the pair of FSO heads reduces visibility for a few meters, while clouds as well as haze have a similar impact on FSO systems. Low visibility naturally reduces the efficiency and overall availability of FSO systems, while the low visibility phenomenon usually occurs in a specific case [9,10].

## 2. Hybrid FSO/RF System

Hybrid FSO/RF systems are based on the principle of using a backup line, the most common radio frequency (RF) line. In case the probability of failure of the primary optical FSO line is high, the communication is switched to the base RF line (in case of insufficiently transparent atmospheric channel) [11].

The principle of switching the FSO link to the back-up RF link is to set a fixed critical boundary of the received RSSI optical output at which the optical transmission of the FSO link is switched to the slow RF channel. When applying such a system, there must be a bad link system between the pair of FSO heads to ensure smooth communication transfer on both sides of the line. Data transmission is thus still realized by only one of the lines. The aim is to ensure the longest possible operation of communications in the primary optical FSO link mode, while maintaining the standard of high availability and quality of communication link [12].

The experimental optical fiber-free system consists of a pair of FSO heads and a monitoring device. The main task of this device, consisting of a Raspberry Pi3 Model B minicomputer and sensors, is to monitor weather conditions along the atmospheric canal. The measurement results are used as inputs for machine-learning methods, for processing and prediction of the received optical power, which is the implementation of RSSI parameter in hard-switching of hybrid FSO/RF line, (Figure 1).

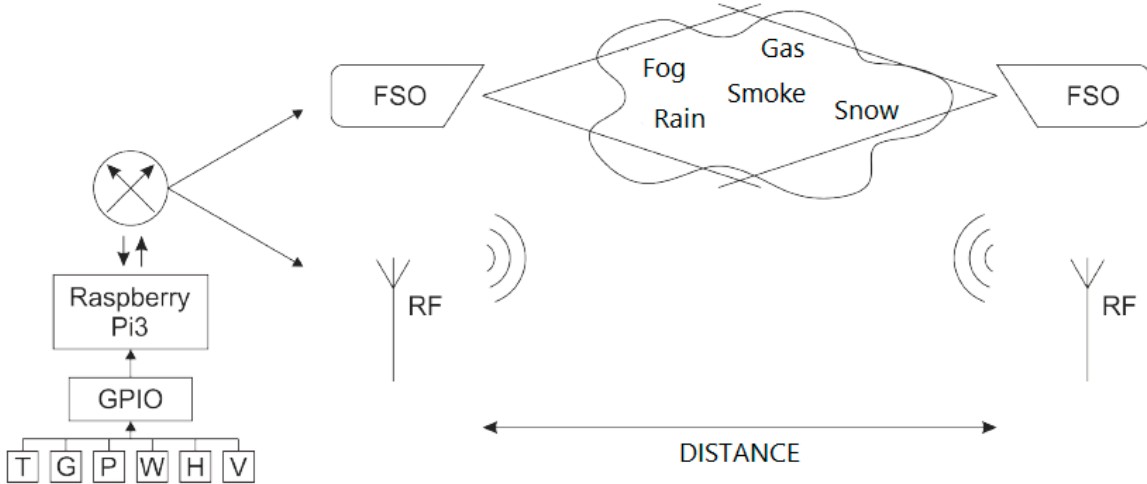

**Figure 1.** Scheme of the hybrid free space optics (FSO)/ radio-frequency (RF) system.

An important question in the design of any monitoring device is the way of collecting, processing and backing up data obtained from regular measurements. The main advantages of the chosen microcomputer include dimensions, price, open operating system, general-purpose input/output (GPIO) bus (possibility of connecting analog and digital sensors), all of which are implemented in the proposed experimental system to analyze the availability and reliability of FSO systems (Figure 2). Monitoring weather conditions along the transmission path between two FSO heads is a key task in the process of determining and evaluating the reliability as well as the availability of FSO systems [13].

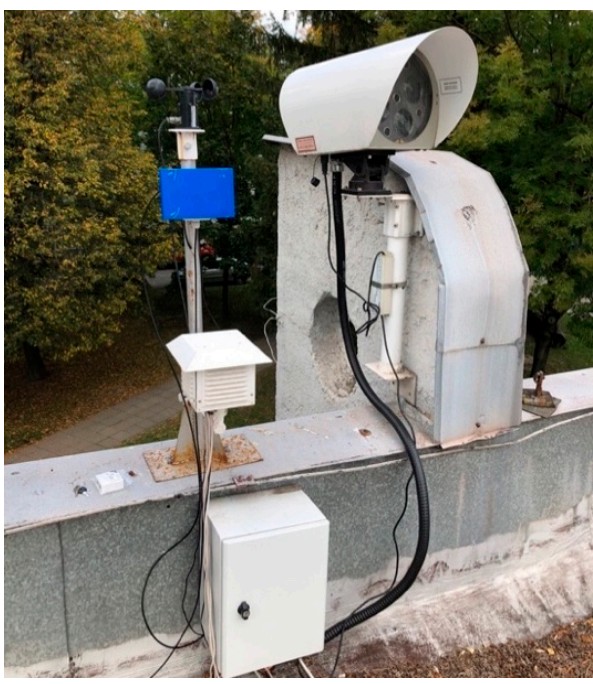

**Figure 2.** Monitoring station.

Visibility measurement based on the backscattering principle is performed in the proposed system using a mini optical fiber sensor (miniOFS). The sensor measures the amount of water droplets (fog particles) in the air that limit visibility [14].

The importance of measuring and monitoring the temperature in the immediate vicinity of the FSO transmitter/receiver, as well as along the transmission path of the FSO line, is related to atmospheric

turbulence. Atmospheric turbulence is characterized as an accidental phenomenon caused by a sudden change in temperature and pressure in the atmosphere along the transmission path between two FSO beams. As a result of such weather developments, the randomly formed clusters vary in size. A large number of small optical prisms or lenses are formed, which ultimately help to create constructive or destructive interferences [15].

Monitoring and detection of the concentration of fog particles is realized by means of a simple optical sensor. Measuring the change in the density (concentration) of particles in the atmosphere can make a significant contribution to solving the problem of early backup of the FSO line by a backup line and thus ensure the high availability of the optical link.

Measurement of wind speed in the process of analysis of the availability of FSO systems is important, especially in terms of identifying the occurrence of optical turbulence in the vicinity of the transmission path between the FSO transmitter and the receiver. The main adverse effects of atmospheric turbulence on the optical signal are scintillation [16].

Obtaining actual humidity data provides valuable information in the process of analyzing the availability of FSO systems when the humidity is approaching 100%. The humidity value of 100% is critical during the formation of the mist, which has the greatest negative contribution to the overall atmosphere of the transmitted light beam [17].

The time evolution of the barometric pressure value is significantly related to the air temperature. Together with other observed concomitant phenomena in the atmosphere, air pressure influences the formation of atmospheric turbulence, which negatively affects the transmitted optical signal. Atmospheric turbulence makes up one quarter of the total contribution of the propagating optical beam [18].

## 3. Depth Data Analysis

The amount of RSSI optical power received (dB/km) is an important decision indicator in the process of hard-switching the hybrid FSO/RF system. Modern FSO/RF systems switch to the RF transmission mode after exceeding the critical low received optical power limit on the receiving side. This information must be distributed and synchronized on both sides of the link. Back to optical mode, the hybrid system will switch if the current RSSI value is above the critical level for a certain amount of time. The operating frequencies of commercially deployed RF lines are typically around 60 GHz (unlicensed bandwidth), which implies that the transmission rate of such a system is disproportionately lower than that of an optical link [19,20]. There is an effort to develop a reliable prediction of RSSI parameter development, based on which it would be possible more accurately identify the necessary time of communication of the hybrid system in RF mode.

Effective reduction of the hybrid system operating time in RF mode will increase the maximum utilization time of the high transmission speed of the FSO link, while maintaining the condition of continuous communication between two nodes of the hybrid FSO/RF lines. The optical beam emitted from the FSO head is naturally attenuated by passing through the atmospheric channel, depending on the current nature of the weather conditions. The received optical power RSSI detected on the optical receiver can be interpreted as a function of a set of parameters describing the nature of the weather conditions. In the present system, parameters are measured and recorded in real time: temperature, humidity, particle concentration in air, visibility, barometric pressure and wind speed (Table 1). Based on the results of theoretical analysis of atmospheric channel for FSO line, a monitoring system for selected physical quantities received optical power (RSSI), average temperature ($T$), average pressure ($P$), average temperature from sensor DHT22 ($T22$), average humidity from sensor DHT22 ($H22$), average temperature from sensor ds18b20 ($Temp$), average wind speed in m/s ($W_{speed}$), average wind speed in volts from sensor ($W_{volt}$), average dust in the air from sensor gGM3G gp2y1010au0f ($G_{mgm3}$), average visibility from miniOFS sensor ($Vis$).

**Table 1.** Example of data set preprocessing.

| Timestamp | RSSI | *T* | *P* | *T22* | *H22* | *Temp* | *W*$_{speed}$ | *W*$_{volt.}$ | *G*$_{mgm3}$ | *Vis* |
|---|---|---|---|---|---|---|---|---|---|---|
| 21 September 2019 07:51 | −39.66 | 17.40 | 990.40 | 15.40 | 49.09 | 11.12 | 0.48 | 0.39 | 12.0 | 4000 |
| 21 September 2019 07:52 | −32.37 | 17.30 | 990.52 | 15.40 | 49.09 | 11.18 | 0.40 | 0.39 | 13.0 | 4000 |
| 21 September 2019 07:53 | −25.00 | 17.35 | 990.50 | 15.55 | 49.19 | 11.21 | 0.41 | 0.39 | 13.0 | 4000 |
| 21 September 2019 07:54 | −25.00 | 17.55 | 990.52 | 15.40 | 48.95 | 11.28 | 0.41 | 0.39 | 11.5 | 4000 |
| 21 September 2019 07:55 | −25.00 | 17.55 | 990.56 | 15.55 | 48.89 | 11.34 | 0.41 | 0.39 | 12.0 | 4000 |

The main software components of the software system to increase the availability and reliability of FSO systems include, in addition to control scripts, a MySQL database, which serves to storage of collected data. From the point of view of backing up the file system, a repository was created on one of the standard servers in case of a minicomputer failure, which serves as an identical image of the minicomputer's directories. In addition to backing up the directory system, a Slave MySQL database is located on this server.

The main aim of this analysis is to make the most accurate prediction of the RSSI output parameter and thus the *y* matrix from the *X* input matrix data. The correlation results are shown in Table 2. In the next step, based on the results from Table 2, a basic matrix of input parameters *X* was created. This matrix consists of pressure—*P*, temperature—*T*, particle concentration parameters in air—*G*, visibility—*V*, air humidity—*H* and wind speed—*W*.

$$
X = \begin{bmatrix}
P_1 & T_1 & G_1 & V_n & H_n & \cdots & W_1 \\
P_2 & T_2 & G_2 & V_n & H_n & \cdots & W_2 \\
\cdots & \cdots & \cdots & \cdots & \cdots & \cdots & \cdots \\
P_n & T_n & G_n & V_n & H_n & \cdots & W_n
\end{bmatrix}
\tag{1}
$$

**Table 2.** Correlation of input parameter with output parameter received signal strength identifier (RSSI).

| Parameter $X_i$ | Correlation Coefficient ($X_i$, RSSI) |
|---|---|
| Received optical power | 1.000000 |
| Average temperature | 0.405682 |
| Average pressure | 0.033820 |
| Average temperature DHT22 | 0.382871 |
| Average humidity DHT22 | −0.044555 |
| Average temperature ds18b20 | 0.467381 |
| Average windspeed WU anemometer | 0.041365 |
| Average wind voltage WU anemometer | 0.041365 |
| Average dust gp2y1010au0f | 0.459923 |
| Average visibility miniOFS | 0.428446 |

The whole data set of input variables *X* with corresponding values of the output variable *y* was randomly divided into two parts in a ratio of 80:20. The ratio was chosen according to the standard practice of machine-learning methods depending on the size of the data file [21].

$$
y = \begin{bmatrix} y_{RSSI.1}, & y_{RSSI.2}, & \cdots, & y_{RSSI.n} \end{bmatrix}^T
\tag{2}
$$

More than 80% of the set of data training cases were used in the model training process. The remaining 20% of the test data were used to verify the accuracy of the timed model. In this analysis, the scikit-learn library was used to compare the individual machine-learning algorithms on the data collected. The matrix of input variables *X* consists of the data from Table 3. The data for the matrix of the output variable *y* is obtained from Table 4. Time alignment of data from both tables based on the *TimeStamp* column created a matrix of cases *P*. The data set contains records from three years, which represents more than 600,000 data.

**Table 3.** Examples of input variables for *X*.

| T P bmp183 | T H dht22 | T ds18b20 | W U Anemometer | G | AAV mOFS |
|---|---|---|---|---|---|
| t: 8.50; p: 990.91 | t: 6.80; h: 95.83 | t: 1.625000 | w: 0.415297; u: 0.397262 | g: 9 | vis = 3333 |
| t: 8.40; p: 990.88 | t: 6.70; h: 95.699997 | t: 1.625000 | w: 0.415297; u: 0.397262 | g: 10 | vis = 3333 |
| t: 8.40; p: 990.87 | t: 7.00; h: 95.92 | t: 1.625000 | w: 0.411430; u: 0.397137 | g: 9 | vis = 3333 |
| t: 8.40; p: 990.95 | t: 6.70; h: 96.699997 | t: 1.687000 | w: 0.848435; u: 0.411263 | g: 11 | vis = 3333 |
| t: 8.50; p: 991.04 | t: 6.90; h: 96.33 | t: 1.687000 | w: 0.867772; u: 0.411888 | g: 9 | vis = 3333 |

**Table 4.** Examples of output variables for *y*.

| ID Record | TimeStamp | RSSI |
|---|---|---|
| 195,085 | 2 February 2019 20:10:26 | −31 |
| 195,084 | 2 February 2019 20:09:26 | −31 |
| 195,083 | 2 February 2019 20:08:26 | −31 |
| 195,082 | 2 February 2019 20:07:26 | −31 |
| 195,081 | 2 February 2019 20:06:26 | −31 |

Decision-trees are used for classification and regression prediction of target *y* parameter (RSSI). Although this mathematical apparatus was introduced more than two decades ago, the potential and massive deployment in data depth analysis has only been in the last few years. Decision-trees are relatively simple to interpret and have an illustrative learning process. A decision-tree can be defined as a structure that uses the principle of partitioning data into smaller data set using decision rules to split a large data set [22].

Methods of creating decision-tree algorithms are often called inductive learning algorithms [23]. The training system maps and searches the rules space and selects the rules that most precisely classify data training set. The rules represent a generalization of the training cases on the specific data set that the system was acquainted with. Trees are in decision-tree theory generated by the so-called Top Down Induction of Decision-trees (TDIDT) [24]. There must be a training set at the beginning of the decision-tree generation process. The training set consists of a set of cases $Q = \{Q_1, Q_2,..., Q_n\}$. Individual cases are defined by a tuple of test features, the steps being input variables $X = \{X_1, X_2,..., X_k\}$. These can be either numerically or verbally. In the present FSO/RF system, only numeric values are used. A set of cases must contain the values of the output variable *y* in addition to the input variables. The general training set of cases has the following structure.

$$Q = \begin{bmatrix} x_{11} & x_{12} & \ldots & x_{1k} & y_1 \\ x_{21} & x_{22} & \ldots & x_{2k} & y_2 \\ \ldots & \ldots & \ldots & \ldots & \ldots \\ x_{n1} & x_{n2} & \ldots & x_{nk} & y_n \end{bmatrix} \tag{3}$$

When creating the decision-tree, the input training set is divided into smaller subsets, which gradually characterize the values of the output variables. The recursive division process is performed until the termination condition is met. In the process of atmospheric channel analysis for FSO/RF system an extensive system of cases of input variables *X* with the corresponding output variables *y* was designed. The output variable file *y* in this case is RSSI. The first of the group of algorithms applied to the set of *Q* cases is the decision-tree regression algorithm [24,25]. The matrix of cases *Q*, as input for the mentioned model, which can be formulated as follows:

$$Q = \begin{bmatrix} ID & P & T & G & V & H & W & RSSI \\ 33,023 & 998.24 & 3.387 & 11.0 & 4000.0 & \cdots & \cdots & -26 \\ 109,253 & 995.56 & -1.875 & 12.0 & 3658.0 & \cdots & \cdots & -25 \\ 65,342 & 972.36 & 6.458 & 12.5 & 3658.0 & \cdots & \cdots & -26 \\ 15,773 & 990.045 & 14.254 & 12.5 & 3658.0 & \cdots & \cdots & -24 \\ \vdots & \vdots & \vdots & \vdots & \vdots & \vdots & \vdots & \vdots \end{bmatrix} \Big\} 80\% \tag{4}$$

The basic input parameter for the decision-tree regression algorithm is *max depth* or *depth of regression*. The assessment of the credibility of the timed model is carried out by several methods. Mean square error and so-called score or coefficient of determination were used in this analysis [26].

$$mse(y, \hat{y}) = \frac{1}{n} \sum_{n}^{n-1} (y_i - \hat{y}_i)^2 \qquad (5)$$

where $y_i$ is the actual value of the predicted parameter, $\hat{y}_i$ is the predicted value and $n$ is number of samples. The best achievable *mse* value is 0. The score parameter prescription is as follows

$$R^2(y, \hat{y}) = 1 - \frac{\sum_{i}^{n-1}(y_i - \hat{y}_i)^2}{\sum_{i=0}^{n-1}(y_i - \overline{y}_i)^2} \qquad (6)$$

where $\overline{y}_i$ represents the mean value of the Receive Signal Strength Indicator expressed as

$$\overline{y} = \frac{1}{n} \sum_{n}^{n-1} y_i \qquad (7)$$

## 4. Results

The determination coefficient—respectively the *score*—determines the probability of how well the learning model will predict will be the samples of the output variable *y*. The best achievable score value is 1. The *max depth* parameter is selected based on the optimum *mse* and *score* values. For the selected algorithm, the *max depth* parameter was sequentially tested from 5 with step 1 to 300. In Figure 3 shows the development of the mass and *score* depending on the *max depth*.

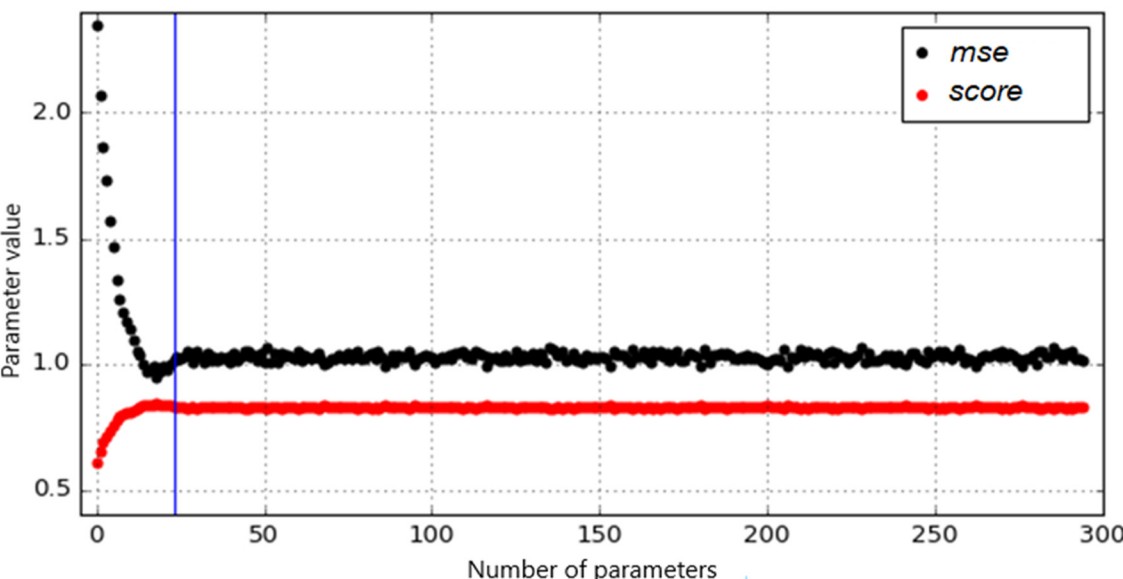

**Figure 3.** Specifics of the max depth parameter for the decision-tree regression method.

The cyclic training model showed that the optimum value of the parameter *max depth* for *mse* = 0.94 and *score* = 0.84. Next Figure 4 illustrates the comparison between predicted (red color) and real (black color) RSSI values.

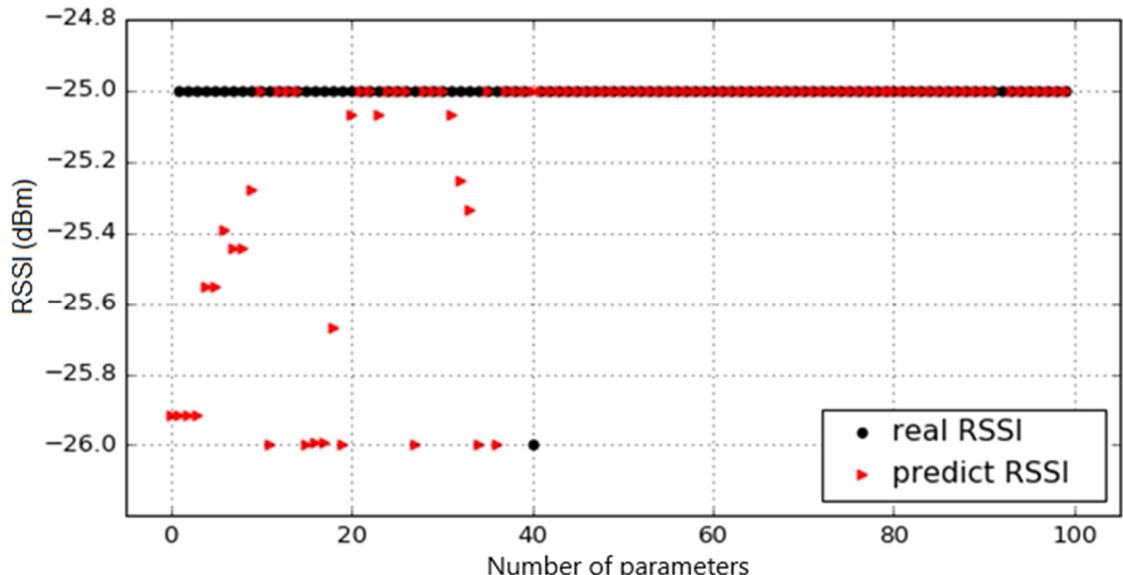

**Figure 4.** Example of the decision-tree regression method prediction.

*4.1. Decision-tree Regression with Adaboost Regressor Method*

Technically, the combination of decision-tree regression algorithms with AdaBoost regressor is considered a random forest method and generally improves the *mse* and *score* values. The main difference from the previous approach was the fixed *max depth* setting of 23, with the *n-estimators* parameter changing between 100 and 700 in steps of 10. The results of the training process with respect to the size of the *mse* and the *score* parameter are shown in Figure 5.

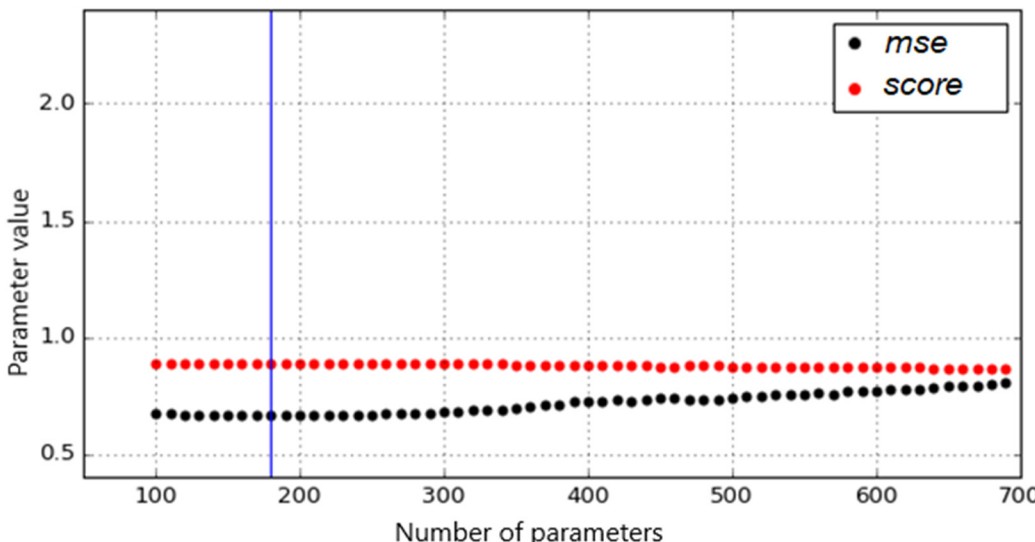

**Figure 5.** Determination of *n-estimators* parameter for decision-tree regression using AdaBoost regressor.

It was shown that for a *max depth* of 23 and the number of *n-estimators* 180 trees, it is possible to reduce the mean quadratic error of the *mse* to 0.66 and the *score* to 0.89. An example of comparison of predicted and real RSSI values is shown in Figure 6.

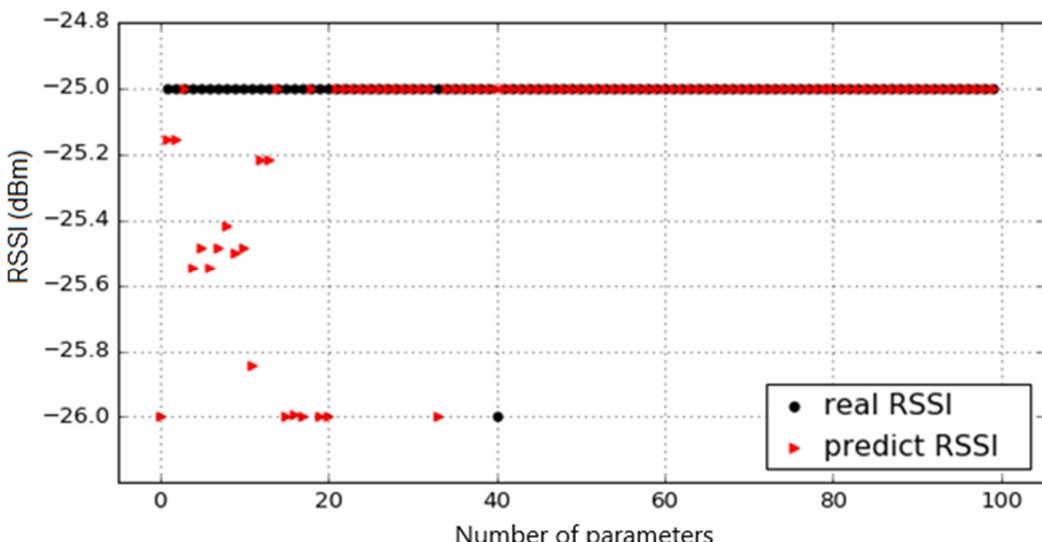

**Figure 6.** Example of prediction of the decision-tree regression method with AdaBoost regressor.

*4.2. Random Forest Regression Method*

The random forest regression method was applied to the case matrix with a gradual change of the *n-estimators* parameter in the interval from 100 to 2000 with step 50. The maximum depth of the individual *max depth* tree was fixed to 23. From the comparison in Figure 7 shows that the optimum value of *n-estimators* is 800, which is indicated by the blue vertical line that defines the optimum values for the *mse* and *score* parameters.

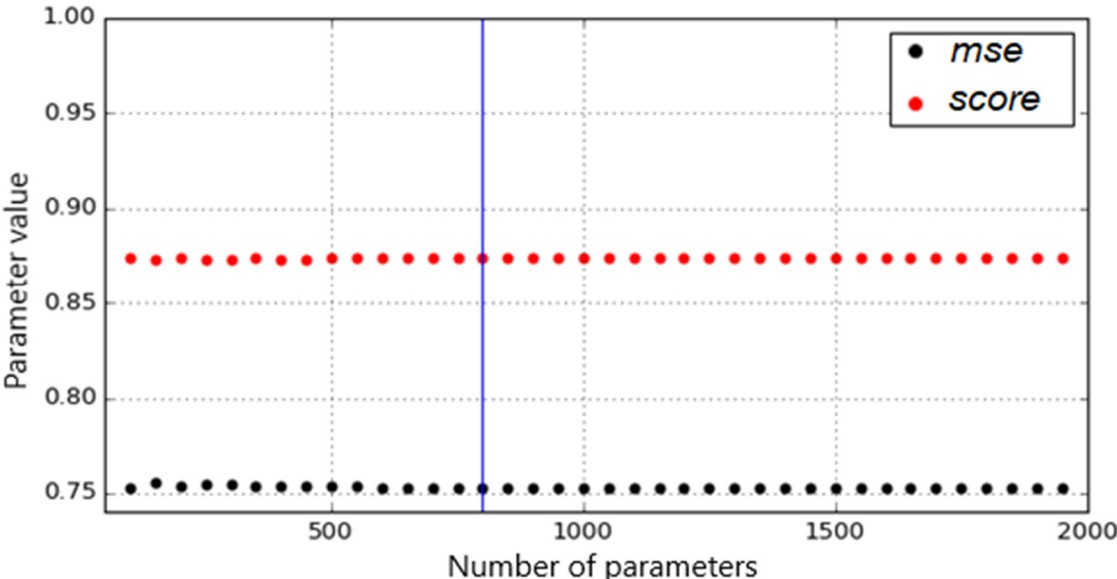

**Figure 7.** Determination of the *n-estimators* parameter using the random forest regressor method.

The achieved *mse* value is 0.75 and the *score* value is 0.87. An illustration of the immediate prediction of the first 100 samples from the test matrix is shown in Figure 8.

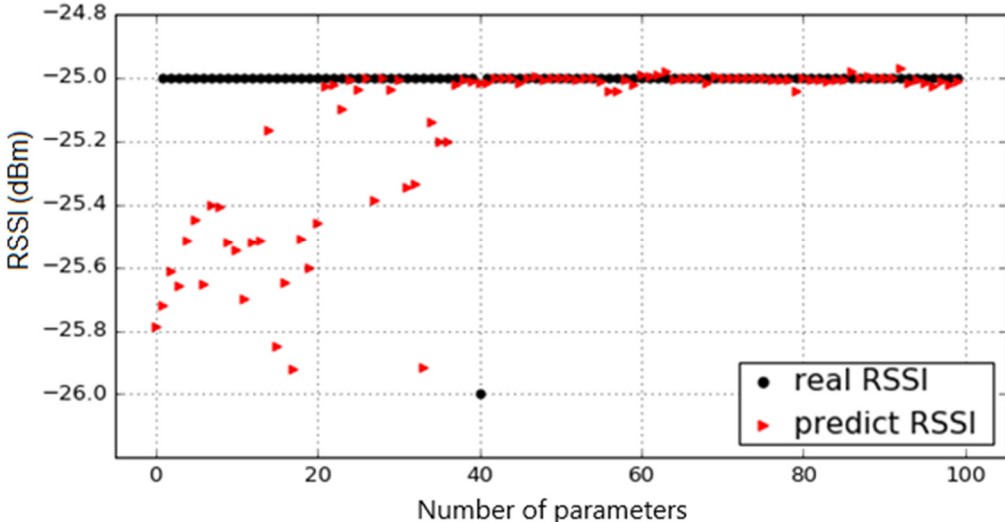

**Figure 8.** Example of a prediction using the random forest regressor method.

### 4.3. Gradient-Boosting Regressor Method

Similar to the previous random forest regression method, the case matrix was subjected to a case-by-case test of the optimum number of trees *n-estimators* [26]. Various combinations of input parameters were tested, while again the number of trees gradually changed from 100 to 2000 with step 50. The results of the training are shown in Figures 9 and 10.

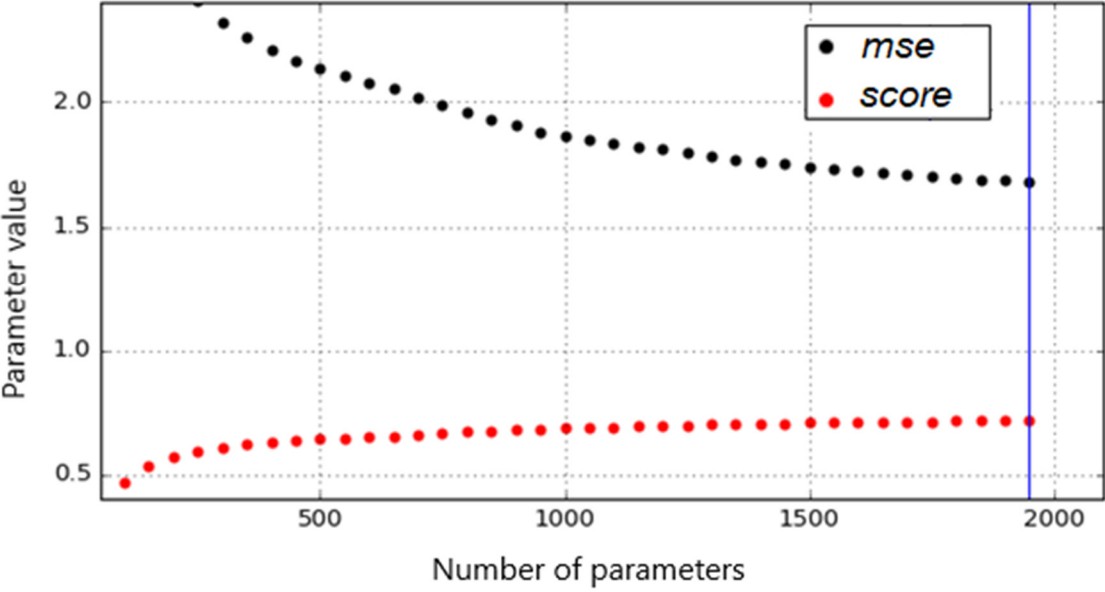

**Figure 9.** Determination of the n-estimators parameter using the gradient-boosting regressor method.

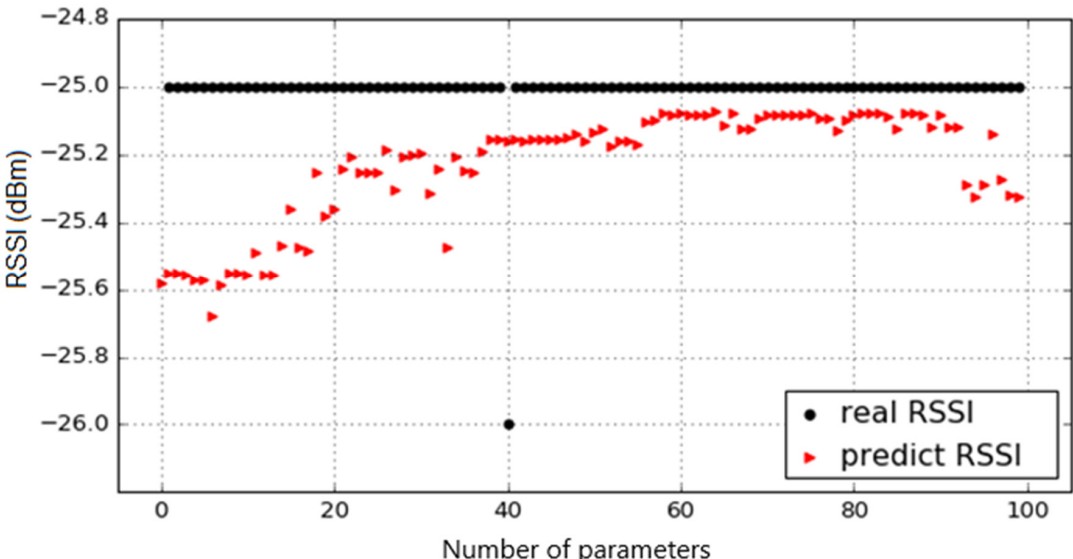

**Figure 10.** Example of prediction using gradient-boosting regressor.

Instant diagnosis and prediction of the received optical power in real time is difficult to implement from the application point of view, and in fact is not sufficiently effective. The delays that arise when measuring individual parameters, storing them in a database and processing them for a certain ~90 s, make it more objectively to predict the RSSI parameter a few minutes ahead. The prediction image of RSSI development is to prevent unnecessary switching of the link to the base line and back, respectively unjustified persistence of the communication line in the RF mode. At this point, two basic tasks arise. The first is to correctly design the case matrix of the *P* cases, and the second is to optimally select a training model that will adequately describe most of the possible scenarios of the case matrix (Figure 11).

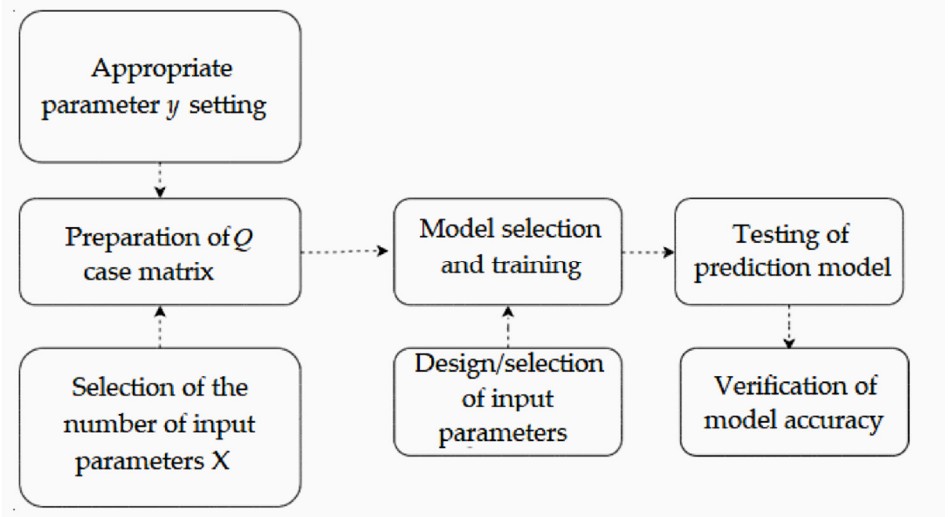

**Figure 11.** Design of training process of machine-learning models.

The matrix of input parameters *X* is significantly extended in the process of time prediction by data gradually shifted over time. An input parameter matrix, such as an input for a training model, is no longer made up of data arranged in a logical timeline sequence. On the contrary, the actual measured data are neglected due to the time delay of the measurement and processing process. Use data suitably time-shifted back by −*x* min as shown in Equation (8). An important part of preprocessing the *X* shift data input matrix is the optimal selection of the number of input parameter shifts over time. With each

offset, the parameter input matrix is expanded by 6 new columns $P_{1-xmin}$, $T_{1-xmin}$, $G_{1-xmin}$, $V_{1-xmin}$, $H_{1-xmin}$, $W_{1-xmin}$, thus increasing the model calculation time as well as to greater time consuming in the prediction itself. The next step in the data preprocessing process is to shift the output parameter matrix. The easiest approach is to move the output parameter $y_{shift}$ forward one discrete value.

$$y_{shift} = \begin{bmatrix} y_{1-xmin,} & y_{2-xmin,} & \cdots, & y_{n-xmin} \end{bmatrix}^T \tag{8}$$

RSSI prediction was made during the night with very foggy and variable weather patterns. Three different pre-trained models were used in the prediction process. For the AdaBoost regression model in combination with decision-tree regression, the following combination of input parameters *n-estimators* = 130 and maximum depth of individual tree *max depth* = 10 was selected based on previous analysis. Figure 12 shows the predictive output of this model in red. The second of these models was the random forest regressor. The input parameters for this model were selected as follows: The maximum number of *n-estimators trees* = 300 and the depth of the individual tree *max depth* = 16. By comparing the training and test data groups, *mse* and *score* were calculated at 0.698 and 0.88. The resulting prediction is represented by blue in Figure 12. The last third model analyzed is gradient-boosting regressor, whose input training parameters were chosen as in the first model, *n-estimators* = 130 and *max depth* = 10.

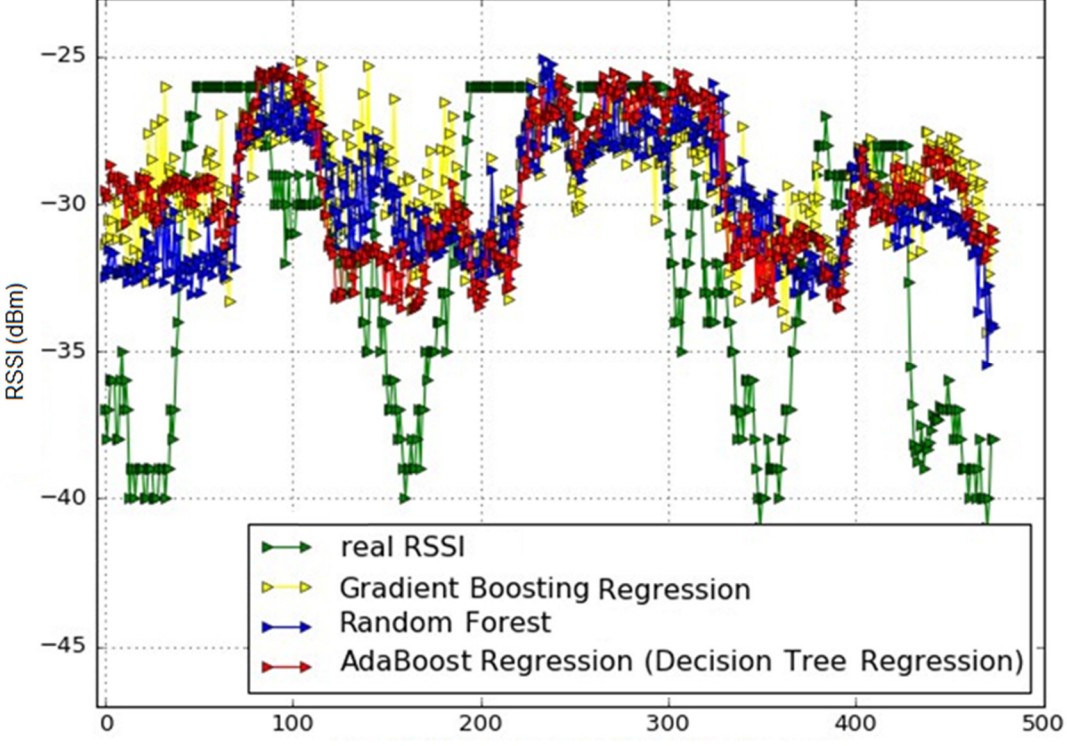

**Figure 12.** Comparison of prediction models.

Model training for the regression methods of the random forest and gradient-boosting random forests shows that the optimal values of the number of trees in the random forest are in the order of hundreds of n estimators. However, such a result proved to be incorrect during the practical verification of the proposed models and is proof of the pre-estimation of the so-called overfit training process. AdaBoost regression with decision-tree regression proved to be the most suitable method. The main advantage of regression prediction lies in the exact determination of the predicted value of the RSSI parameter, in contrast to the classification, which divides the examined data set into groups or classes (in this case it is an RSSI parameter).

## 5. Conclusions

Reliable RSSI prediction is critically dependent on several important steps. In this study, an in-depth regression analysis of data was analyzed. Model training requires the construction of an adequate parameter input matrix, as well as the choice of machine-learning model parameters (depth of individual tree *max depth* and number of trees in *n-estimators* forest). Qualitative indicators of adequacy of the proposed model are *score* parameters, whose value should ideally be equal to one and *mse*, whose value should converge as much as possible to 0. Based on these two criteria, they were classifiers calculated optimal as values of *max depth* and *n-estimators*. Gradual change and evaluation of the *max depth* parameter showed that its optimum value is 23. Model training for random forest and gradient-boosting regression methods show that the optimum values for the number of trees in the random forest are hundreds of *n-estimators*.

However, such a result proved to be incorrect in practical verification of the proposed models and is evidence of pre-estimation of the so-called overfit training process. The main advantage of the regression prediction lies in the accurate determination of the predicted value of the RSSI parameter. The correct design of the in-depth data analysis model for prediction is based on correct design of the input data matrix *X*. Two types of input matrices *X* were designed for regression analysis. To ensure sufficient information richness of the input parameter matrix consisting not only of the currently measured data of the weather parameters, they are also supplemented with data from the past. Thus, the first type of parameter input matrix consists of 30 columns. This approach created a time vector sequence of data from individual monitored parameters, which also includes the trend curve of the development of a particular parameter. The trained models of regression classifiers were verified in real time on data at a time of rapid weather changes.

**Author Contributions:** R.H. and P.Š. wrote the study. Ľ.O. and S.M. proposed the original idea. J.P. analyzed the data and Ľ.D. checked the study. All authors have read and agreed to the published version of the manuscript.

**Funding:** This work was founded by the Slovak Research and Development Agency (no. "APVV-17–0208)-Resilient mobile networks for content delivery".

**Conflicts of Interest:** The authors declare no conflicts of interest.

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
