# Peer review of "Prediction of Received Optical Power for Switching Hybrid FSO/RF System"

_electronics, doi:10.3390/electronics9081261_

Round 1

Reviewer 1 Report

The authors presented an interesting practical problem and applied appropriate machine learning algorithms. However, the article has some weaknesses that need to be corrected.

  • The introduction lacks the current state of the literature on the subject under consideration. It is not exactly described what is the aim of the article. Nothing is mentioned about the proposed estimation methods.
  • Some important shortcuts are not explained, e.g. RF, GPIO.
  • In the data analysis there is no information about the size of the data, why such and not other parameters were taken into account? What is the impact and importance of these parameters - which are more important and which are less important in the prediction process?
  • The presented results are not very clear. There is no suggestion from the Authors as to which method is recommended in the problem under consideration. Besides, the names of parameters overlap, e.g. "P" means both training set and pressure. There are no descriptions in the diagrams (it is about axles).
  • Currently the literature on the methods used is very extensive. However, there are only a few items quoted in the article. It should be enriched.

Reviewer 2 Report

Please indicate the purpose, a method and novelty on introduction specifically.

The estimated precision at the time of low RSSI looks low as a result of figure 12. Please show me this reason and improvement method.

Reviewer 3 Report

This articles addresses Free Space Optics (FSO) systems. My comments are as follow:

  1. It is not clear why prediction would be needed, at least as it is mentioned right within the title (Prediction of Received Optical ...). This idea is also iterated in the Conclusions section, i.e. "Reliable RSSI prediction is critically dependent on several important steps." The authors are invited to debate more over this aspect.
  2. A simple check on IEEExplore platform or others (Elsevier, Springer, etc) returns interesting titles for articles, most of them addressing implementations. There are hundreds of papers, if not thousands.
  3. The number of references is not correlated with the real number of pertinent references published in the literature. The authors are invited to justify their research idea with more relevant articles, related to their work and principle. 12 references does not look so convincing, taking into account that some of them are general.
  4. Introduction is too long and contains many basic information.
  5. Figure 1: it is not clear how fog, gas, smoke etc. could be quantified, how they could affect communications, at least from theoretical perspective. In addition, this picture is not complete. It would be interesting to the reader to see the entire transmission link and distinct link characterization depending on the channel parameters and model.
  6. How about the complexity of the prediction system?

Round 2

Reviewer 2 Report

I judge that places pointed out is corrected appropriately.

Reviewer 3 Report

1. disadvantages -> drawbacks

2. "RF communication is degraded by heavy rain" -> "heavy" is true for higher frequencies (tens of GHz, especially at 60 GHz and > 100 GHz)

3. Fig. 1 is not clear, please use vector graphics, it is not so complicated anyway. In addition, figures 3-12 should be processed similarly.

4. "In Figure 3 shows the development" !?

5. "The wavelengths of FSO lines fall within the unlicensed frequency spectrum." -> Is there any standard regulating such communications within unlicensed spectrum? ISM represents unlicensed spectrum but we have standards developed in those bands (0.4 GHz, 0.9 GHz, 2.4 GHz etc).
